# Individual Differences in Digital Game-Based Supply Chains Management Learning: Evidence from Higher Vocational Education in Taiwan

Ya-Cing Jhan [1], Pin Luarn [2] and Hong-Wen Lin [1,*]

1 Department of International Trade, Chinese Culture University, Taipei 11114, Taiwan; kelly6824@hotmail.com
2 Department of Business Administration, National Taiwan University of Science and Technology, Taipei 10607, Taiwan; luarn@mail.ntust.edu.tw
* Correspondence: woodylin34@hotmail.com

**Abstract:** This study aims to investigate the differences in the learning path and value between students of different genders and educational backgrounds in digital game-based learning in Taiwan's higher vocational education. A game-based digital Supply Chains Learning System was used to explore the value network based on "system attributes—learning consequences—target value" from the perspective of the student. To investigate the research question, this study conducted a cross-analysis of differences in gender and education background. Findings from the study revealed that irrespective of gender and educational background in higher vocational education in Taiwan, students developed distinctively different learning paths, values, and emphasis from specific system attributes. These findings will not only enable trainers and educators to learn more about the differences in learning by individuals but will also serve as useful guidelines for the improvement of the teaching strategy used by digital games developers.

**Keywords:** individual differences; supply chains learning system; digital game-based learning; means-end chains theory; higher vocational education





## 1. Introduction

### 1.1. Educational Environment and Background

In response to the rapid development of Taiwan's industrial structure and societal needs, the core of Taiwan's vocational education has been set to equip talents with the ability of practice and innovation, with a focus on information technology (IT), problem-solving, and interdisciplinary integration. Through secondary education and practice, vocational talents become the important navigators for promoting economic development and industrial research and development. Therefore, vocational education has been playing an important role in Taiwan's economic development. Taiwan's vocational education is divided into three levels: Senior high vocational schools, junior colleges, and institutions/universities of technology. Senior high vocational schools aim to develop base-level technicians. Junior colleges are set to cultivate intermediate technologists. Institutions/universities of technology are established to provide higher vocational education for the research and development of high-tech and cultivation of advanced professionals [1]. While higher education is popular in Taiwan, emphasis on integration of theory and practice is the focus of most vocational schools. Due to some restrictions (such as insufficient teaching space, lack of teachers with practical experience, or teaching using outdated technology, etc.), some schools cannot even provide an actual work environment for students to practice. As a result, school education cannot effectively connect with industrial needs. To narrow the education-to-employment gap, the Ministry of Education in Taiwan has been proactively studied the implementation plans of vocational education in Germany and the USA [2,3], hoping to develop the instructor expertise development policy from successful foreign

examples to help vocational instructors enrich their subject knowledge and improve their teaching quality [4]. More flexible and innovative curricula should be planned, and teaching methods should be developed for vocational education in order to deal with future industrial needs [1].

The rapid development of technology has not only given birth to more resources and new ways of learning, such as interactive e-books [5,6], educational simulation games [7,8], learning systems [9], Massive Open Online Courses (MOOCs) [10] etc., but has also helped students cultivate faster logical thinking and information processing [11]. Training for business management now also incorporates information technology as an assistive tool [12]. This helps trainees to better understand the methodologies and theories of management while allowing them to refine their practical skills through, for instance, business simulation games [8]. Business simulation games create realistic simulations of real-world scenarios and serve as tools that can be used to help students make more informed and better decisions [8]. Studies have shown that business simulation games offer significant benefits and help students with decision-making and improve other management-related skills and competence [13]. Games can also provide experiences with a specific management approach for the mitigation of potential risk and loss from poor decision-making in real life. Hence, students can use them in workplaces [14].

### 1.2. Business Management Learning

Global supply chain and logistics management play a crucial role in sustainable development and the key competitive advantage for corporations [15,16]. In Taiwan, the higher vocational education system hopes to focus on raising talents with professional skills instead of focusing on academic theory like college. Therefore, industry-academia cooperation, internships, and simulation games are included in the supply chain management course or management course in vocational high school (university of technology). Students can improve competence in different learning environments before joining the workforce [17]. For instance, Ferguson and Drake [18] introduced the risk management of the supply chain by discussing the current shortage of the global supply chain for tissue paper due to COVID-19. Song et al. [19] integrated stock shortage, supply chain competition, and supply distribution due to COVID-19 into the simulation game to teach a supply chain course. These two examples show that these topics help learners cope with supply chain issues when joining the workforce. Previous studies confirm that higher vocational education increases employability [20].

The Bullwhip effect, caused by the asymmetry of information between different suppliers [21], is still the greatest challenge to supply chain management [22,23]. This results in an escalation and distortion of consumer demand for upstream vendors at different levels of a supply chain. It causes dramatic inflation of estimated demand, which is then met by an actual distribution that is far more modest than the projections [24]. To strengthen corporate competitiveness, previous studies started to propose a scientific model for supply chains in different industries to conduct decision management [25] and evaluate the performance of the supply chain process [26]. In an attempt to understand the importance of supply chain management during school for students, Sterman adapted an experiential game—the Beer Distribution Game in 1989, to demonstrate what really happens in a supply chain. In the game, students took on the roles of four suppliers at different levels of a supply chain, this gave them a very clear idea of the decisions that had to be made by suppliers at each level [27]. It is important to note that elements, such as market competition, price fluctuation, and unanticipated cancelation of orders were excluded from the Beer Game to provide a simpler supply chain simulation [28]. A less difficult gaming environment was found to be more helpful to students without management education backgrounds and allowed a better understanding and experience of the nuances of supply chain management and the Bullwhip effect.

### 1.3. Individual Learning Differences

The intellectual message processing of individuals from different age groups and with different backgrounds differs when learning different types of knowledge [29]. Previous studies of the differences between the learning processes in individual students have indicated that gender played a significant role in the learning process. This was notable in the methods used to learn [30], the approach to information processing [31], and the use of information technology [32]. Dennis et al. [32] maintained that male students proved to be better able and more active in the use of information technology and the resolution of encountered problems and issues than their female counterparts. They also demonstrated a higher degree of acceptance for the use of technology. Incidentally, there are also studies that suggest females had greater capacity in the diverse use of information technology to accomplish tasks that are more sophisticated [33]. Not only that, students of different genders also exhibited significant differences in terms of psychological response and feelings [34]. Female learners tended to emphasize interpersonal relationships and communication skills [34], social interaction [35], and also preferred to share their information [36]. In contrast, male students prioritized the accomplishment of given objectives over everything else [34]. Chung and Chang [37] pointed out that most male students preferred the content of digital-game-based learning to be intense, while female learners preferred that it be mellow in nature. Garber Jr. et al. [38] further pointed out that male serious game participants prefer the learning mode abstract conceptualizer, and female serious game participants prefer the learning mode concrete experience. On the other hand, students of different educational backgrounds also showed substantial differences in terms of their learning preferences, learning strategies, and cooperative learning [39–41]. Students with Natural or Pure Science backgrounds gravitated towards the cultivation of imagination and creativity and were more open to embracing new ways of learning [42]. In contrast, students from the Humanities and Social Sciences focused more on the identification and solution of problems [43]. In their study, Lam et al. [39] also argued that students with Natural Science backgrounds perceived computers as a type of learning resource while students of the Humanities and Social Sciences treated computers as a classroom tool. They also found that students in the latter group were more active in their use of computers for learning than those in the first group. Teachers are, therefore, advised to apply different teaching models [42] and approach to learning outcome assessment [44] when working with students of different educational backgrounds to further improve learning efficacy.

Realistically speaking, empirical results on the differences in learning for individual students are already available in specific publications. However, very few studies have investigated the goals and values pursued in the learning process from the student's point of view. Vocational educators in Taiwan should be equipped with innovative thinking and the ability of practice and interdisciplinary integration. More specifically, the purpose of this study is to compare the differences in the learning paths and values for students of different genders and educational backgrounds in a cross-analysis of the two groups. To our knowledge, this study is the first to discern the differences in the learning path and value of students of different genders and backgrounds in digital game-based learning to investigate the area-specific learning differences to develop policies for cultivating talents with interdisciplinary integration ability in the future.

## 2. Theoretical Background

### 2.1. Digital Game-Based Learning

Numerous studies have shown that when elements of quests, multimedia, interactivity, and scenario simulation are incorporated, digital game-based learning (DGBL) can be more effective compared to a traditional teaching format in boosting student motivation [45–48] and learning outcome [49,50]. Not only that, missions in educational games also help students to develop their capabilities for problem-solving and organization of new knowledge [51]. Empirical studies have also shown that in lessons that are prepared and presented

in an online environment, students assigned to the DGBL experimental groups demonstrated greater learning motivation, flow, and positive learning attitude. DGBL prompted students to actively participate and enjoy the process of learning [52]. This learning method not only helps students to engage in in-depth thinking [53] but also helps to enhance focus and self-awareness [54]. Nevertheless, the difficulty and sophistication of design for DGBL are higher compared to those for average games that are created for entertainment and recreational purposes. Should the DGBL designers fail to achieve seamless integration of teaching content and learning objectives in a game [55], students will engage in the game without being able to reach the deeper levels of thinking and introspection that are highly desirable [56].

More and more teachers involved in business management and administration training are including Business Simulation Games (BSGs) as a part of their standard curriculum and approach [57,58]. Examples of such courses and training include management sciences [59], information systems [60], and supply chain management [61]. Arquero et al. [62] and Fito-Bertran et al. [50] believe that language competence, computer and techno-logical literacy, decision-making, problem-solving, teamwork, and leadership are essential skills that students in business management must acquire. All of which can be achieved through the use of BSGs. Previous studies have shown that students have expressed high levels of satisfaction and acceptance for BSGs [8,63] because they were not only able to attain concrete learning outcomes through the process of playing the games but were also inspired to further their exploration and learning in the discipline.

However, the design of BSGs in recent years has gradually drifted from the improvement of learning to a heavy emphasis on drawing student attention [64]. There is more focus on the aspect of classroom learning and a failure to incorporate proper teaching design [65]. Although many studies support the integration of BSGs technology and vouch for its effectiveness in offering versatility and realistic learning experience, BSGs do not necessarily guarantee effective learning [66,67]. On top of that, most studies and assessments of the outcomes of DGBL have been made from the perspective of the teacher. Very little research has been done that includes reviews of the paths and values of learning from the students' point of view. In the past, the role of the teacher was often that of an entitled authority figure with the power to determine the method as well as the student learning path. However, contemporary studies have shown that when a teacher takes the role of a facilitator, motivator, guider, collaborator, adviser, and moderator of the learning process, student progress is much better [14,68].

### 2.2. Means-End Chains Theory

Means-End Chains (MECs) theory (as shown in Figure 1) is an effective technique that captures the structure of perception and the process of its formation in the mind of a person pertaining to specific tiers of product information [69]. MECs are primarily used to examine the outcome of feelings and ultimate personal value that people acquire through specific attributes of a product or service after they have used it [70]. MECs also serve as an ideal approach in an exploration of the experiences and processes of learning that take place within the minds of students of different genders and educational backgrounds.

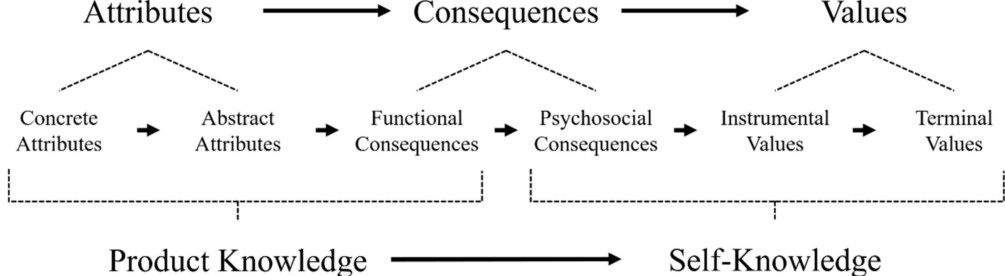

**Figure 1.** Means-End Chains Model.

MECs were developed by Gutman [70] after he integrated the theoretical frameworks proposed by Howard [71], Rokeach [72], Vinson et al. [73], Young and Feigin [74]. MECs are essentially a structure that comprises three primary elements: Attributes, Consequences, and Values [75,76]. The purpose of MEC is to link the "Consequences" and "Values" that users may derive from the "Attributes" of a specific product or service and use the chain to explain the reason that led the users to make a choice. "Means" refers to consumer perception and preferences for a tangible or intangible product, service, or sentiment, which are defined as "Attributes", while "End" encompasses the benefits or consequences a consumer might construct from these "Attributes" to satisfy their "Value".

Attributes are the basis of MECs, and they are the perceived characteristics that consumers associate with their preferred products or services [77,78]. In other words, an "Attribute" is something that can be felt and can be either tangible or intangible [79,80]. Not only that, it can be further separated into "Concrete Attributes" that refer to tangible aspects [81,82] and "Abstract Attributes" that refer to intangible aspects [82–84].

Consequences are on the second MECs tier and can be considered as the impact consumers derive, either directly or indirectly, from the use of a product or service. Consequences are a connection between Attributes and Values [85]. Consequences can include desired benefits as well as risks that a user would wish to avoid [84]. They can be further separated into "Functional Consequences", which result in more specific or direct user experience, such as the use of multimedia instruction to draw the attention and make teaching more effective [86], and the abstract "Psychological Consequences" that refer to the perceived mindset of the user [87]. This could mean that digital learning takes individual differences in students into account and can, therefore, ensure their learning rights and privileges [86].

Value resides at the highest level in the theoretical framework of MECs and it represents the ultimate state of existence longed for by all people [70]. It is the manifestation of abstract perception or something at an even higher level than an individual seeks to attain through specific attributes [88]. A value is determined strictly by a subjective experience and can be further separated into "instrumental values" (the preferences or perception of a behavior) and "terminal values" (the ultimate state of existence that all people desire to achieve) [70,72].

MECs have gradually received recognition as a tool for the exploration of digital game-based learning and general gaming. This includes business operation simulation systems [89], social education [90], digital educational games [43], key functionalities of digital learning systems [86], MOOCs [91], MMORPG studies [92], and general digital gaming [93,94]. In this study, MECs have been chosen for the exploration of the learning paths students take and the structure of "system attributes—learning consequences—target value". The learning path is a successive process based on MECs structure. Each layer can trigger the next layer through its feature. For instance, system attributes trigger learning consequences, and learning consequences trigger target value. The primary aim of the present study is to identify and discern individual differences in playing a BSGs, as well as any learning differences, that might exist between male and female students and those with different educational backgrounds.

## 3. Research Methodology

### 3.1. Material

In the early 1960s, a card-based simulation game that involved supply chain management and was known as "the beer game" was developed at the MIT Sloan School of Management. The beer game operates on the principle of supply and demand for beer as a commodity [95] with the goal of helping students understand the Bullwhip effect [96,97]. Many teachers and educators have since used the beer game, or taken it as the core component, in the design of a PC game [96–98] that gives students a better understanding of supply chain management flow and provides experience in sophisticated decision-making [97].

In this study, Supply Chains Learning System (SCLS) was chosen as the experimental tool. SCLS is a supply chain simulation system that is based on the beer game and each chain in SCLS has four major items or roles: Factory, Distributor, Wholesaler, and Retailer. Each role has to be taken by players who operate and manage the supply chain. Teachers assign roles based on the number of students involved. A role may be assigned to a group or to one individual. The teacher can also configure the number of supply chains with which the students are required to work. SCLS offers cooperative learning, role-playing, and competitive learning. The preliminary settings for the system environment, which are made by the teacher, require the determination of the actual consumer demand for each round and the number of rounds to be played. Other parameters that can be changed include the latency between order placement and cargo shipping and the availability of specific operating information, such as volume of distribution, the inventory of downstream vendors, on-order inventories, etc. Players can swap roles between positions high up on the supply chain or further down. This helps them gain firsthand experience of the impact of information opacity. Students playing the game will learn that every decision they make involves the concept of cost and that poor decisions can lead to either a shortage of products or excessive inventory. Both result in additional unit cost in the form of shortage or excess. When the game reaches the final round, the costs for each role are tallied and supply chains that ran on lower costs reflect superior operational performance.

### 3.2. Procedure

To construct the learning path and values of a subject with MECs as a theoretical framework, the study requires some specific analytical processes, Laddering, Content Analysis, Implication Matrix, and Hierarchical Value Map (HVM) [43,99].

The first step involved the application of Laddering, for interviews with subjects who have had previous experience in SCLS, to find their learning path of "system attributes, learning consequences, and target value". Laddering can be hard or soft. Hard laddering is based on structured questioning that enables the rapid elicitation of responses from subjects in a short time and is suited for studies that involve a large sample size [100]. Soft Laddering involves non-structured interviews and is more effective in uncovering abstract feelings and perceptions that exist in deep psychological layers of the mind. Since the aim was the identification of individual differences that exist between the genders and students with different educational backgrounds, the emphasis was on an in-depth understanding of learning paths and values to determine such differences, soft laddering was used in this study (see the interview questions in Section 3.3).

After data collection by soft laddering had been completed, the study proceeded to its second phase, content analysis. This involved the conversion of raw, primary data into usable content by systematic and objective description [101]. The attributes, consequences, and values mentioned by the subjects were coded and sorted based on the characteristics of their choice of words and phrases. In coding, the guidelines on stability, reproducibility, and accuracy as proposed by Reynolds and Gutman [99] and Krippendorff [101] were followed. After exhaustive discussion, a set of rules for coding was created and finalized. This served as a framework for the classification of subject input into specific categories of attributes, consequences, and values that were closest in terms of semantics before the analysis categories were named.

The third and final phase of the study involved the processing of coded outcome into an Implication Matrix that was used to compute the number of connections between the elements, such as Attributes-Consequences, Consequences-Consequences, and Consequences-Values before they were represented in the HVM [99]. The study has adopted three important principles for HVM mapping. The first item is the arrow direction. The HVM follows the MECs theory, and the link relationship starts from the left (attributes) and ends on the right (values). The second item is the thickness of the link line. The thicker the link line, the higher the number of links. The third item is the number of links. The number of links between elements is indicated next to the link line. However, if all the links in

the Implication Matrix had to be faithfully represented in the HVM in full, the results would become so involved that they would obscure the critical links that really mattered. Therefore, a crucial cut-off value was established that helped in the determination of links that were stable. The elimination of all links below cut-off made the interpretation of the HVM much more manageable [102].

Specifically, the study has adopted two Cut-off Value setting methods. The first method sets a standard cut-off value and only considers the link relationship exceeding the standard value, which can effectively present the important links. The second is the "top-down cut-off" method, it is a concept that prioritizes the top-1 link relationship. For the cut-off level of top-1, if the link relationship is not obvious or a complete structure cannot be formed, the top-2 should be set as the standard. To ensure that the HVM could be presented in a more concise manner, the cut-off level for gender and educational background analysis was set at 4 to accurately capture the links prioritized by the subjects. A "top-down cut-off" concept as proposed by Leppard et al. [103] was used for the cross-gender and educational background analysis, with the level, for male and female Business Administration students set at top 1. The cut-off level for students of both sexes in Science and Engineering and Digital Learning and Education was set to top 2 to describe the paths that mattered most to them.

### 3.3. Sampling and Data Collection

Because the selection of suitable samples for this study involved restrictions, the snowball sampling method was chosen for data acquisition. Lin and Tu [89] pointed out that there should be consistency between the research material and the games used when studies on digital game-based learning are carried out. For this reason, only persons who had previous experience of playing the SCLS chosen for this study were eligible subjects. This prevented potential discrepancy in perception and bias in the research findings. Since an in-depth interview method had been used as the method of sample collection, the study could be regarded as qualitative. Others have noted that the focus of qualitative research ought to rest on versatility and depth [104] and in this study, emphasis has been placed on the quality of samples, not quantity. Reynolds et al. [78] maintained that a reliable study should feature a subject pool of no less than 20 individuals, while Leppard et al. [103] suggested a sample size of 50–60, when it came to interviews, would lend greater credibility and make findings more representative. The subjects selected for this study were graduate students currently enrolled in specific programs of Business Administration, Science and Engineering, and Digital Learning and Education at Taiwan's University of Science and Technology. For each program, 20 subjects (10 male and 10 female) were chosen. They were between the ages 23 and 28, and a total of 60 individuals took part in the three programs.

In-depth interviews require a great deal of stamina and energy to ensure that all the valuable data can be collected while they are still fresh in the minds of the subjects. The interviews were kept to between 30–45 min in duration as far as possible. The following structure was used in all the questions used in the interviews: (1) What attributes of SCLS attracted you or have left a lasting impression?, (2) what consequences do you expect from these attributes?, and (3) what internal values or feelings do such consequences satisfy? These questions were raised sequentially and repeatedly to stimulate recall of their SCLS experiences while the interviewer guided them in deep thought about the abstract and concrete consequences of the event before a reflection on the values the experience had enabled them to fulfill. The interview was ended once the subject was unable to think of any further attributes of significance.

## 4. Data Analysis and Results

### 4.1. Coding

A List of Values (LOV) as proposed by Kahle [105] was used to rank the values in this study. The LOV framework used was developed based on the Maslow Hierarchy of Needs and the Value Survey created by Rokeach [72]. The advantage of LOV as a framework

lies in the fact that it relates closely to day-to-day life and requires no revision as time passes. Furthermore, the LOV theory has been a common choice of technique in the past MECs-related studies. As such, the study has adopted LOV as the basis of compiling SCLS learners' target values. In particular, nine important value definitions are available in LOV, including sense of accomplishment, self-fulfillment, fun and enjoyment in life, security, warm relationships with others, self-respect, sense of belonging, excitement, and well-respected. A total of thirty-one elements have been identified, with nine system attributes, thirteen learning consequences, and nine target values (as shown in Table 1), as well as for the presentation of the specific number of times each element is mentioned by the subjects.

### 4.2. Content Analysis Result for Gender

Among the system attributes prioritized by subjects, male subjects perceived Teamwork (A2) to be the most important, followed by Role of supply chain (A1). Incidentally, female subjects also emphasized the significance of Role of supply chain (A1) as an attribute and believed Customizable model variables (A5) to play a vital role in SCLS.

With regards to the learning consequences derived from system attributes, it was found that from the thirteen learning consequences identified, three had zero mention by male subjects: Casual and burden free (C9), Reduces instruction load (C11), and Helps learners to grasp the situation (C13). In contrast, the female subjects mentioned all thirteen learning consequences. This suggests that the male subjects were more focused on the specific learning consequences they were looking for, while the female subjects were more diverse in their choices. Among the consequences mentioned, both male and female subjects believed that both Train organizational thinking (C5) and Improve operational performance (C2) provided by SCLS were important. The male subjects made more mention of Incorporation of real scenarios (C1), Facilitate cooperation and interaction (C3), and Experience bullwhip effect (C12) compared to their female counterparts. The female subjects made more mention of Boosts learning results (C4) and Inspire competitive mentality (C10) compared to their male peers. This shows that besides the key learning consequences for SCLS, male and female subjects focused on different secondary learning consequences.

Individual subjects, both male and female, acknowledged Sense of accomplishment (V1) and Excitement (V8) to be the target values they would have expected from playing SCLS. The only difference between the subjects was the fact that men claimed to benefit from the additional value of Self-fulfillment (V2) while women named Fun and enjoyment of life (V3) as a bonus value.

### 4.3. Content Analysis Result for Education Background

Naturally, subjects from different educational backgrounds also prioritized different system attributes. However, the sole exception was Role of supply chain (A1) that was identified by subjects from all three educational backgrounds. Students in Business Administration emphasized Customizable model variables (A5), whereas those in Science and Engineering and Digital Learning and Education focused more on Teamwork (A2). On top of that, Digital Learning and Education students also identified Simple operating interface (A8) to be a key system attribute.

**Table 1.** Item codes of SCLS data.

| Item | Total | Genders | | Education Backgrounds | | | Genders and Educational Backgrounds | | | | | |
| --- | --- | --- | --- | --- | --- | --- | --- | --- | --- | --- | --- | --- |
| | | | | | | | Business Administration | | Science and Engineering | | Digital Learning and Education | |
| | | Male | Female | Business Administration | Science and Engineering | Digital Learning and Education | Male | Female | Male | Female | Male | Female |
| System attributes | 171 | 82 | 89 | 48 | 64 | 59 | 23 | 25 | 35 | 29 | 24 | 35 |
| A1 Role of supply chain | 44 | 21 | 23 | 20 | 12 | 12 | 11 | 9 | 5 | 7 | 5 | 7 |
| A2 Teamwork | 38 | 22 | 16 | 2 | 18 | 18 | 1 | 1 | 14 | 4 | 7 | 11 |
| A3 Operation statement | 14 | 4 | 10 | 2 | 9 | 3 | 1 | 1 | 3 | 6 | 0 | 3 |
| A4 Provide information | 13 | 9 | 4 | 1 | 6 | 6 | 1 | 0 | 4 | 2 | 4 | 2 |
| A5 Customizable model variables | 28 | 11 | 17 | 18 | 7 | 3 | 8 | 10 | 2 | 5 | 1 | 2 |
| A6 Pre-lesson overview | 6 | 3 | 3 | 0 | 3 | 2 | 0 | 0 | 1 | 2 | 2 | 1 |
| A7 Computer-assisted instruction | 7 | 2 | 5 | 0 | 3 | 4 | 0 | 0 | 2 | 1 | 0 | 4 |
| A8 Simple operating interface | 16 | 8 | 8 | 1 | 5 | 10 | 0 | 1 | 3 | 2 | 5 | 5 |
| A9 Presentation in tables and diagrams | 5 | 2 | 3 | 4 | 1 | 0 | 1 | 3 | 1 | 0 | 0 | 0 |
| Learning consequences | 273 | 139 | 134 | 71 | 101 | 101 | 38 | 33 | 57 | 44 | 44 | 57 |
| C1 Incorporation of real scenarios | 30 | 20 | 10 | 12 | 7 | 11 | 10 | 2 | 4 | 3 | 6 | 5 |
| C2 Improve operational performance | 41 | 20 | 21 | 16 | 14 | 11 | 7 | 9 | 8 | 6 | 5 | 6 |
| C3 Facilitate cooperation and interaction | 28 | 17 | 11 | 6 | 11 | 11 | 3 | 3 | 9 | 2 | 5 | 6 |
| C4 Boosts learning results | 15 | 5 | 10 | 3 | 6 | 6 | 1 | 2 | 2 | 4 | 2 | 4 |
| C5 Train organizational thinking | 64 | 32 | 32 | 16 | 24 | 24 | 8 | 8 | 14 | 10 | 10 | 14 |
| C6 Boosts motivation to play | 5 | 3 | 2 | 0 | 2 | 3 | 0 | 0 | 2 | 0 | 1 | 2 |
| C7 Innovative and fun | 8 | 5 | 3 | 4 | 2 | 2 | 3 | 1 | 1 | 1 | 1 | 1 |
| C8 Helps learners to get into the scenario | 19 | 9 | 10 | 2 | 7 | 10 | 0 | 2 | 3 | 4 | 6 | 4 |
| C9 Casual and burden free | 5 | 0 | 5 | 1 | 1 | 3 | 0 | 1 | 0 | 1 | 0 | 3 |
| C10 Inspire competitive mentality | 23 | 10 | 13 | 5 | 8 | 10 | 1 | 4 | 5 | 3 | 4 | 6 |
| C11 Reduces instruction load | 2 | 0 | 2 | 0 | 0 | 2 | 0 | 0 | 0 | 0 | 0 | 2 |
| C12 Experience bullwhip effect | 30 | 18 | 12 | 6 | 16 | 8 | 5 | 1 | 9 | 7 | 4 | 4 |
| C13 Helps learners to grasp the situation | 3 | 0 | 3 | 0 | 3 | 0 | 0 | 0 | 0 | 3 | 0 | 0 |
| Target value | 171 | 82 | 89 | 48 | 64 | 59 | 23 | 25 | 35 | 29 | 24 | 35 |
| V1 Sense of accomplishment | 63 | 28 | 35 | 19 | 20 | 24 | 9 | 10 | 8 | 12 | 11 | 13 |
| V2 Self-fulfillment | 20 | 11 | 9 | 4 | 8 | 8 | 2 | 2 | 4 | 4 | 5 | 3 |
| V3 Fun and enjoyment of life | 19 | 8 | 11 | 9 | 5 | 5 | 4 | 5 | 4 | 1 | 0 | 5 |
| V4 Security | 17 | 9 | 8 | 5 | 10 | 2 | 3 | 2 | 6 | 4 | 0 | 2 |
| V5 Warm relationships with others | 11 | 4 | 7 | 2 | 7 | 2 | 0 | 2 | 4 | 3 | 0 | 2 |
| V6 Self-respect | 8 | 3 | 5 | 1 | 1 | 6 | 0 | 1 | 0 | 1 | 3 | 3 |
| V7 Sense of belonging | 7 | 5 | 2 | 3 | 3 | 1 | 3 | 0 | 2 | 1 | 0 | 1 |
| V8 Excitement | 22 | 11 | 11 | 5 | 7 | 10 | 2 | 3 | 5 | 2 | 4 | 6 |
| V9 Well-respected | 4 | 3 | 1 | 0 | 3 | 1 | 0 | 0 | 2 | 1 | 1 | 0 |

In terms of similarity between desired learning consequences, students from all three educational backgrounds perceived both Train organizational thinking (C5) and Improve operational performance (C2) to be vital consequences of learning. As for where they varied, Business Administration students maintained the Incorporation of real scenarios (C1) from SCLS to be crucial, while Science and Engineering students felt the Experience bullwhip effect (C12) to be a significant learning consequence. Interestingly, the study also found that Digital Learning and Education students were more extensive in terms of the learning consequences they thought to be desirable. Apart from the common learning consequences associated with students from the other two groups, they also pointed to Incorporation of real scenarios (C1), Facilitate cooperation and interaction (C3), Helps learners to get into the scenario (C8) and Inspire competitive mentality (C10) to be significant learning consequences.

Students from all three groups agreed that the Sense of accomplishment (V1) that SCLS offered was a target value. However, they did differ in their choice of the target value next on the ladder of importance. Business Administration students named Fun and enjoyment of life (V3) as their second most important target value, while Security (V4) and Excitement (V8) were chosen by Science and Engineering, Digital Learning and Education students respectively.

### 4.4. Content Analysis Result for Cross-Gender and Educational Background

As mentioned previously, in addition to looking at the subjects in isolation, the study has also made cross-analysis of individual subjects of different gender and educational background. In terms of system attributes, both male and female students from Business Administration prioritized Role of supply chain (A1) and Customizable model variables (A5). Male students from Science and Engineering pointed to Teamwork (A2) as the most important system attribute while the choices by female Science and Engineering students were more diverse as to what they believed to be important system attributes. These included Role of supply chain (A1), Operation statement (A3), and Customizable model variables (A5). Digital Learning and Education students of both sexes chose Role of supply chain (A1), Teamwork (A2), and Simple operating interface (A8) as the key attributes. Interestingly, female Digital Learning and Education students also confessed that the attributes of Operation statement (A3) and Computer-assisted instruction (A7) of SCLS were desirable for them. This was an element that their male counterparts did not bring up.

It was also found that as far as similarities in the desired learning consequences for subjects in the cross-gender and educational background analysis were concerned, apart from specific elements, subjects of different genders (but of the same educational background) did not show significant differences in terms of their choices. Both groups identified Improve operational performance (C2) and Train organizational thinking (C5) to be the learning consequences they would prioritize. Science and Engineering subjects, both male and female, believed Experience bullwhip effect (C12) to be the most crucial attribute while Digital Learning and Education students of both genders further identified Incorporation of real scenarios (C1) and Facilitate cooperation and interaction (C3) to be material. With regards to their differences in the choice of learning consequences, male Business Administration students placed heavier emphasis on Incorporation of real scenarios (C1), and male Science and Engineering students found Facilitate cooperation and interaction (C3) to be more important. In contrast, male Digital Learning and Education students picked Helps learners to get into the scenario (C8), while their female counterparts favored Inspire competitive mentality (C10).

As for target value, most subjects, despite their differences in gender and educational background, chose Sense of accomplishment (V1). Business Administration students of both sexes also acknowledged Fun and enjoyment of life (V3) as an additional target value, while Science and Engineering students of both sexes wanted the extra value of Security (V4). It is also worth mentioning that male Digital Learning and Education students were

more focused on target values and did not mention other values, such as Fun and enjoyment of life (V3), Security (V4), Warm relationships with others (V5), and Sense of belonging (V7). In contrast, female Digital Learning and Education students mentioned all target values that had been identified in the study, with the sole exception of Well-respected (V9).

*4.5. HVM Analysis Result for Gender*

Analysis of gender HVM as shown in Figure 2. Three specific attributes were linked to Train organizational thinking (C5) in the learning consequences for male students, as well as being linked to two other attributes. This makes Train organizational thinking (C5) an attribute of critical importance that was also a functional node in the learning path. The male students believed the Customizable model variables (A5) system attribute provided a different competitive model environment, while Role of supply chain (A1) offered different roles with a varied approach to management. The characteristics of diverse simulation derived from these two system attributes enabled male students to achieve the consequences of Incorporation of real scenarios (C1) and Train organizational thinking (C5) in their SCLS experience. Incidentally, past studies showed that DGBL can be used to improve flow experience and learning outcome for undergraduate students [106] and Incorporation of real scenarios (C1) happens to be one of the key elements that generate flow experience. On the other hand, male students placed extra emphasis on the system attribute of Provide information (A4) and it stands out as the greatest difference in a comparison of subjects of different genders. The fact is, SCLS Provide information (A4) gives players access to information, such as inventory, demand, costs, etc. From the changes in this information, male subjects became cognizant of the changes that took place in terms of the roles they played in the simulation. In other words, from the consequence of Experience bullwhip effect (C12), they were not only able to derive the value of Sense of accomplishment (V1) but also to arrive at the learning consequence of Train organizational thinking (C5). From there, the male subjects developed two branching paths of learning. One led directly to the target value of Self-fulfillment (V2), while the other went to the consequence of Improve operational performance (C2) before reaching the target value of Sense of accomplishment (V1).

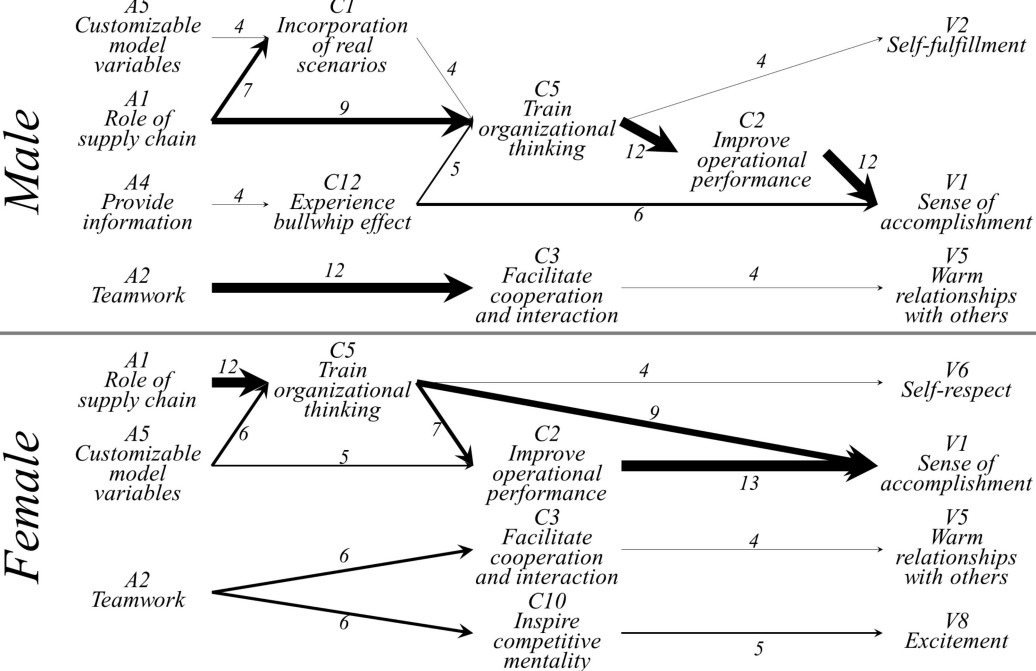

**Figure 2.** Analysis of Gender HVM (Cut-off Level: 4).

While the female subjects also emphasized both Role of supply chain (A1) and Customizable model variables (A5), they differed from the male subjects in the fact that they did not arrive at the consequence of Incorporation of real scenarios (C1) from the two system attributes. Interestingly, past studies found that although female students have a lower inclination to engage in learning through digital learning games than male students, their learning outcome from the activity was superior to that of their male counterparts [107]. They focused more on the aspect of the content and knowledge that the digital learning games had to offer [108]. It seemed that the female subjects sought to arrive at the learning consequence of Train organizational thinking (C5) through the two system attributes, and from this, they derived the target values of Sense of accomplishment (V1) and Self-respect (V6).

The focus on Teamwork (A2) in SCLS was the same for both the male and female subjects, in that they believed that every role in the supply chain was responsible for Facilitate cooperation and interaction (C3), and this would lead to the target value of Warm relationships with others (V5). In addition, female subjects also arrived at the additional learning consequence of Inspire competitive mentality (C10) from Teamwork (A2) and the value of Excitement from their exposure to a competitive atmosphere. In a nutshell, male students engage in DGBL with greater emphasis on consequences that relate to socializing, while female students are oriented more towards the completion of the given tasks.

*4.6. HVM Analysis Result for Educational Background*

Analysis of education background HVM as shown in Figure 3. As far as Business Administration students were concerned, the fact that SCLS offered four different roles that required them to play with different management strategies, as well as the attribute of Role of supply chain (A1), meant that they had to resort to thinking in ways that facilitated careful planning and organization. In addition, teachers would be able to make the decision-making scenarios more diverse and sophisticated using the SCLS attribute Customizable model variables (A5), given that Business Administration students had already expressed a wish for adjustment of specific model parameters to match the content being taught. This would not only improve engagement in versatile learning, but also explain why Role of supply chain (A1) and Customizable model variables (A5) had been linked to Train organizational thinking (C5). This would facilitate systematic thought and decision-making and enable students to further Improve operational performance (C2), which leads to the target value of Self-fulfillment (V2) for Business Administration students. The study found that from the learning consequence of Improve operational performance (C2), the path for students of both Science and Engineering and Digital Learning and Education led to Sense of accomplishment (V1) and is significantly different from that of Business Administration students. Besides, the Target value gained by students receiving Digital learning and education, compared to students in other fields. It could be that students receiving Digital learning and education acquired profession associated with digital education and they mostly choose teaching careers. It is the same for the study by Sun et al. [86]. In this study, the teacher used the e-learning system to acquire Self-fulfillment (V2) and a Sense of accomplishment (V1). It is worth discussing that only in-service teachers will get Fun and enjoyment of life when using the e-learning system. Students would not get Fun and enjoyment of life (V3) when using the e-learning system. When designing the educational game system, it is necessary to get a whole picture of the teacher's and student's learning needs. If the system is designed for the one-side user, learners would lose interest or faith in the learning system, such as that the teacher can fully monitor the student's learning progress.

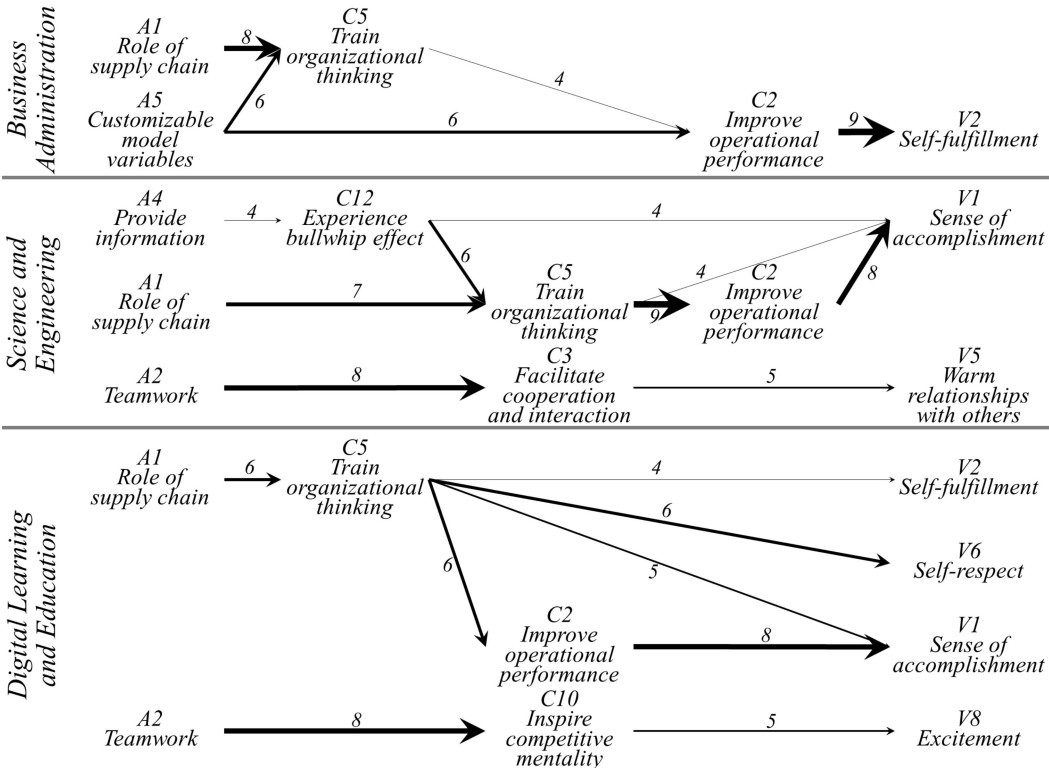

**Figure 3.** Analysis of Educational Background HVM (Cut-off Level: 4).

Compared to students from other educational backgrounds, Science and Engineering students placed a strong priority on the system attribute of Provide information (A4). They believed that management decisions made through the feedback of relevant data not only allowed more science-based management and enabled the Experience bullwhip effect (C12) through SCLS but also made it possible for them to Improve operational performance (C2) through the consequence of Train organizational thinking (C5). Relevant studies have shown that students of Science and Engineering tend to focus on the cultivation of skills and logical thinking. The result of this is they have fewer opportunities to attain a capacity for strategic thinking from the perspective of managers [109]. This means the attribute of Role of supply chain (A1) of SCLS has been perceived by students of Science and Engineering as a vital system attribute for the consequence of Train organizational thinking (C5). These learning consequences have been identified by Science and Engineering students as key sources where they can derive a Sense of accomplishment (V1). In addition, Science and Engineering students also emphasized Teamwork (A2), which facilitates adequate communication and discussion of management decisions that enable them to work together to solve any given issue in the supply chain leading to Facilitate cooperation and interaction (C3), and ultimately to the target value of Warm relationships with others (V5).

As for Digital Learning and Education students, they also believed that the attribute of Role of supply chain (A1) can help to Train organizational thinking (C5). Incidentally, students of Digital Learning and Education felt they could benefit from diverse target values, such as Self-fulfillment (V2), Self-respect (V6), and Sense of accomplishment (V1), as a learning consequence of Train organizational thinking (C5). Students from Digital Learning and Education differed from Science and Engineering students in that they emphasized Teamwork (A2) that involves competition and perceived it as a way to attain great results. Victory in SCLS is attained by the supply chain that manages to run at the lowest cost, this game aspect emphasizes the consequence of Inspire competitive mentality (C10) for students of Digital Learning and Education and offers the target value of Excitement.

### 4.7. HVM Analysis Result for Cross-Gender and Educational Background

Analysis of cross-gender and education background HVM as shown in Figure 4. Students from Business Administration, both male and female prioritized Role of supply chain (A1) and Customizable model variables (A5). However, the men arrived at the consequence of Incorporation of real scenarios (C1) through the two attributes, while the women sought the consequence of Train organizational thinking (C5) through Role of supply chain (A1), which, in conjunction with Customizable model variables (A5), was linked to the consequence of Improve operational performance (C2). In contrast, the men believed that the combination of Role of supply chain (A1) and Customizable model variables (A5) could generate diverse environments that would not only enable them to verify the knowledge they had learned but also help with the Incorporation of real scenarios (C1). From the subsequent flow experience, they were able to arrive at the consequence of Improve operational performance (C2) in the roles they assumed in the supply chain. Robbins and Coulter [110] observed that Management Functions encompassed four specific processes: Planning, Organizing, Leading, and Controlling to efficiently and effectively facilitate the operational target. The male and female Business Administration students both leveraged their knowledge of management functions to achieve outstanding performance, which took them to the target value of Sense of accomplishment (V1) in the end.

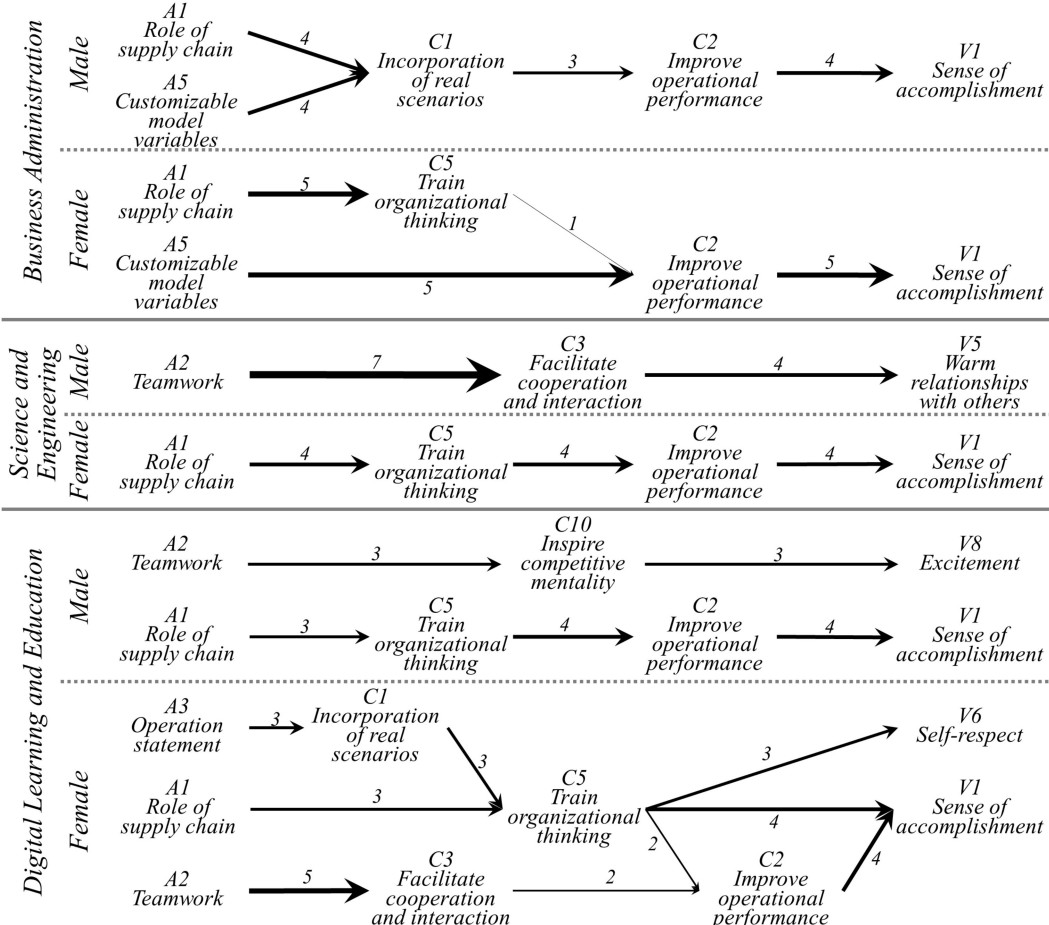

**Figure 4.** Cross-Analysis of Gender and Education Background HVM. Cut-off Level-top1: Business Administration, Science and Engineering; Cut-off Level-top2: Digital Learning and Education.

Students majoring in Science and Engineering, both men and women, varied greatly in the learning paths and values they prioritized. The men utilized Teamwork (A2) to delegate relevant management elements required in the SCLS and assigned specific tasks to different members to make the decision process sounder and more thorough. This led

to the consequence of Facilitate cooperation and interaction (C3), which in turn, brought in the value of Warm relationships with others (V5). Women placed more emphasis on Role of supply chain (A1) as the most crucial key to doing well in SCLS. Their planning of management strategies for each role in the game was careful and deliberate so that regardless of the role each played, delivery of the best possible operational performance was possible. It was also found that female students were able to think about specific processes involving different steps that would expose their management decisions to minimal risk and achieve optimal performance. This meant they could go from the consequence of Train organizational thinking (C5) to Improve operational performance (C2) and ultimately arrive at the value of Sense of accomplishment (V1) as individuals.

The male and female subjects from Digital Learning and Education were found to have developed learning paths that were significantly different in terms of structure. The learning paths of the male students clearly revealed a structure that illustrated a linear influence. The female students, on the other hand, followed learning paths with multiple branching links. In short, the men held the opinion that the characteristics of competitive learning provided by SCLS was crucial. SCLS enables multiple groups of students to compete at the same time and several supply chains can be also run at the same time and the system computes the operating performance for all players at the end of the game. This meant the men believed that Teamwork (A2) discussion in a competitive environment would Inspire competitive mentality (C10) and lead to the target value of Excitement. On the other hand, the Teamwork (A2) on which the women focused on the collective strength that comes from cooperation to Facilitate cooperation and interaction (C3) and to pursue Sense of accomplishment (V1) through the consequence of Improve operational performance (C2). That said, both groups of students sought to Train organizational thinking (C5) through the attribute of Role of supply chain (A1) and reach the target value of Sense of accomplishment (V1) through Improve operational performance (C2). It is worth noting that the female students made an additional link from Train organizational thinking (C5) to the target value of Self-respect (V6). In this study, it was found that when female students played SCLS, they emphasized the importance of systematic thinking and believed the learning consequence of Train organizational thinking (C5) to be vital training that would strengthen their capabilities, and thus offer a sense of Self-respect (V6). Not only that, but female Digital Learning and Education students also brought up a special path of Operation statement (A3) and apparently believed that feedback from it not only helped with the Incorporation of real scenarios (C1) but was also linked to Train organizational thinking (C5) and satisfied the values of Sense of accomplishment (V1) and Well-respected (V9).

## 5. Discussion and Implications

In the study, a SCLS has been chosen as an experimental tool. Thirty-one elements have been identified, containing nine system attributes, thirteen learning consequences, and nine target values. Furthermore, in content analysis results, the system attributes, learning consequences and target values emphasized in different genders, educational backgrounds and gender and educational background cross-analysis are different. The learning consequences acquisition is more concentrated for male students and more diverse for female students. The learning consequences focus by the students participating in digital learning and education is wider. Interestingly, students from different educational backgrounds all believe that Train organizational thinking (C5) and Improve operational performance (C2) are important learning consequences. Finally, in HVM analysis results, studies found significant differences between male and female students, especially in learning consequences and target values. The findings in educational background analysis during SLCS learning show that students with different educational backgrounds focus on different key points. The findings show that there is not much difference between male and female students for System attributes. There is a greater difference in Learning consequences and Target value. This demonstrates that all colleges are on a clear track to fostering talents, but learning differences exist among individuals.

In terms of gender differences, male students emphasized the value of the Warm relationships with others (V5) that Teamwork (A2) delivered for them. It is, therefore, recommended that digital learning software designed with male students in mind should feature online/network connection play functionalities to motivate male students engaged in DGBL and provide an incentive to learn. It should be noted that male students created one additional path Provide information (A4) that the female students ignored. Therefore, to help the men arrive at the value of Sense of accomplishment (V1) through play, relevant tips and information on how specific missions and tasks can be completed should be provided in the game to encourage an interest in playing. Female students preferred to achieve the consequence of Train organizational thinking (C5) through role-playing to complete whatever mission had been given to them, thereby reaching the target value of Sense of accomplishment (V1). As Lowrie and Jorgensen [108] pointed out in their study, female students place heavier emphasis on the aspect of knowledge acquisition when it comes to the content of digital learning games. The study suggests that digital learning software tailored for female students should focus more on the scenario or mission for the role-playing element so that women can leverage their skills for organizational thinking, which would boost their learning outcome and intention.

When using digital games for the purpose of teaching with different student demographics, it is only natural to expect differences in learning behaviors and outcome [64]. Findings from this present study revealed that the key learning paths for students coming from different educational backgrounds showed considerable differences. Business Administration students prioritized a learning path that began from the combination of Role of supply chain (A1) and Customizable model variables (A5) to carry out specific management decisions in game scenarios prepared by the teacher. They sought to arrive at the target value of Self-fulfillment (V2) from Improve operational performance (C2). Findings suggest that digital learning software designed for students of Business Administration should be developed using Scaffolding Theory [111] as the core so that students would be able to build and establish their own game rules. Additionally, teachers could utilize the attribute of Customizable model variables (A5) to create additional tasks, incidents, and contingencies of uncertainty to guide students smoothly through the learning process and help them achieve Self-fulfillment (V2). Lam et al. [39] pointed out that students at business colleges have a tendency to use online communication as a learning tool. It is suggested that software developers include features of online feedback so that teachers and students would be able to interact and provide feedback to other parties. Students from Science and Engineering are primarily concerned with the path that is formed from the attribute of Provide information (A4) than are students from other disciplines. These students believe that the more information relating to the learning objective offered by the gaming platform, the better they will be able to acquire new knowledge. Using SCLS as an example, the consequence of Experience bullwhip effect (C12) leads to Train organizational thinking (C5), which then links to Sense of accomplishment (V1) as a result of learning. Relevant studies have shown that amongst the different models of teaching, heuristic teaching has been proven to be most effective for students in Engineering Colleges [112]. It is, therefore, recommended that developers provide more supplementary materials that are tied to the target learning objective on the software platform. This would not only inspire students to be more proactive in their learning, but also benefit from the value of Sense of accomplishment (V1) once they had actually achieved the objective. Digital Learning and Education students differ from students in other educational backgrounds in the fulfillment of Self-respect (V6) as a target value as a consequence of Train organizational thinking (C5). This might be due to the fact that considering their higher chances of pursuing careers in education, they would be required to act as positive thinking role models for their students. One of the main ways to achieve this is through self-recognition. Therefore, it is suggested that games that are targeted at students who have chosen teaching and education as a profession for some point in the future may do well to incorporate elements of learning

that will help the players cultivate soft skills [113,114], which would allow the learners to fulfill their desire for Self-respect (V6) through the gaming process.

With regards to the differences identified in this study between the genders and by the analysis of educational background, it was found that students with similar educational backgrounds varied only by a small degree with respect to the terms of the system attributes they prioritized. This reflects the fact that schools with academic discipline focus on a clearly defined direction in the provision of knowledge, training, and the nurture of competence in their students. This explains why the system attributes of SCLS that students of different educational backgrounds generally emphasize is closely correlated with their prior knowledge and its application. Nevertheless, it is evident that even with the same system attributes, students of different genders pursued different learning consequences and target values. As pointed out by Lewis et al. [40], for students of different educational backgrounds, something as clear-cut as "collaboration" could carry multiple connotations and meanings. Findings of the present study show that the consequence of Teamwork (A2) as identified by students of different educational backgrounds and the key paths that emerged from the consequence did indeed differ. Male Science and Engineering students emphasized the value of Warm relationships with others (V5) from collaborating with others. It is, therefore, recommended that special abilities or items be assigned to each specific role when it comes to game design so that players could assist one another to complete the intended learning task. However, for male students of Digital Learning and Education, the path from Teamwork (A2) ultimately led to the value of Excitement (V8), and this is why developers should incorporate elements of real-time online high scores/ranking, irregular challenges/elimination matches, etc., to amplify the sense of thrill that is associated with Excitement for such students. On top of that, female Digital Learning and Education students prioritized the learning path of Operation statement (A3) and felt that follow-up feedback from the operating report could help them with Train organizational thinking (C5), and in turn, satisfy their pursuit of Sense of accomplishment (V1) and Self-respect (V6) as target values. It is, therefore, suggested that the software designers include other materials that would supplement the study of the chain in the follow-up feedback. This would help learners to extend their scope and carry on with their pursuit of supply chain management.

## 6. Limitations

This study is based on Taiwan's higher vocational education. Interviewees are graduate students from the age of 23 to 28 studying Business Administration, Science and Engineering, and Digital Learning and Education in Taiwan. The in-service master's program is not included in this study due to the restriction of the study sample. Furthermore, to adapt to the purpose of this study, all the subjects are Taiwanese students. Foreign students are not included.

To extend this study further in future, this study suggests comparing subjects in different countries. A broader educational view can be explored in the higher vocational education delivered in different countries. It would enable researchers to understand the differences among students in different countries learning the supply chain management via SCLS. It is believed that more new knowledge beneficial to differences in individual learning will be acquired from the studies on the effects of differences associated with gender and educational background.

## 7. Conclusions

As established previously, this study presents a report of the results of a cross-analysis of the structure of "system attributes—learning consequences—target value" in three groups of students of both genders with different educational backgrounds who had played SCLS. The value network and learning paths revealed in the investigation offered useful data for teachers interested in the design of digital learning courses and for the developers of relevant software and applications.

This study explores the differences in learning paths for students with different educational backgrounds and genders. It helps higher vocational schools and their teachers plan for adequate educational games and integrate them into the course to assist learners in skill development. Furthermore, past studies found that the students' skill development and professional knowledge and the students' confidence in themselves can be used to predict their career paths [115,116]. In other words, employers want to hire graduates with professional knowledge and skills. These skills must be able to be used the workplace [117]. If the person who designed the vocational education course does not understand the learning differences among students with different educational backgrounds and genders, it is hard to design a digital learning game that meets learner's needs and a high degree of contextualization.

**Author Contributions:** Conceptualization, P.L. and Y.-C.J.; methodology, Y.-C.J. and H.-W.L.; validation, Y.-C.J. and H.-W.L.; formal analysis, H.-W.L.; investigation, Y.-C.J.; data curation, H.-W.L.; writing—original draft preparation, P.L., Y.-C.J. and H.-W.L.; writing—review and editing, H.-W.L. and Y.-C.J.; visualization, H.-W.L.; project administration, P.L. All authors have read and agreed to the published version of the manuscript.

**Funding:** This research was funded by Ministry of Science and Technology in Taiwan (R.O.C.), MOST 107-2420-H-011-001-DR.

**Institutional Review Board Statement:** The study was conducted in accordance with the Declaration of Helsinki, and approved by the Institutional Review Board of University of Taipei in Taiwan (R.O.C.) (Approval Code: IRB-2022-001 and Approval Date: 15 February 2022).

**Informed Consent Statement:** Not applicable.

**Data Availability Statement:** The data presented in this study are available on request from the corresponding author. The data are not publicly available due to the sensitivity of the data.

**Conflicts of Interest:** The authors declare no conflict of interest.

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
