# Peer review of "Individual Differences in Digital Game-Based Supply Chains Management Learning: Evidence from Higher Vocational Education in Taiwan"

_sustainability, doi:10.3390/su14084614_

Round 1

Reviewer 1 Report

The originality and the significance of the research paper should be highlighted in the abstract.

There is lack of clarity within the discussion and findings section. 

The findings of the papers should be clearly highlighted. Limitations are needed to be added. Future directions need to be clarified why they are suggested.

please consider these papers in your manuscript:

Modeling resilient supplier selection criteria in desalination supply chain based on fuzzy DEMATEL and ISM

Maturity evaluation of supply chain procedures by combining SCOR and PST models

Impact of coercive and non-coercive environmental supply chain sustainability drivers on supply chain performance: mediation role of monitoring and collaboration

A Fuzzy Multi-Objective Mathematical Programming Model for a Closed-loop Supply Chain Network Design by Considering Environmental Factors

Design of Closed-Loop Supply Chain Network Considering Uncertainty in the Quality of Recycled Products and Solving It with Lp-Shape Scenario Reduction Algorithm

Author Response

Reviewer 1

Revision Note

The originality and the significance of the research paper should be highlighted in the abstract.

*The study reviewed the abstract and simplified the multiple research objectives. As a result, the originality and significance of the research are more highlighted in the abstract.

There is lack of clarity within the discussion and findings section.

The findings of the papers should be clearly highlighted. Limitations are needed to be added. Future directions need to be clarified why they are suggested.

*Thanks for the advice from the Reviewer. Explanations are added in the “4.8. Summary of Findings” section of the study for clarification.

*“5.2. Limitations and Directions for Future Research” and a description for this section were added at the end of the article.

please consider these papers in your manuscript:

1.Modeling resilient supplier selection criteria in desalination supply chain based on fuzzy DEMATEL and ISM.

2.Maturity evaluation of supply chain procedures by combining SCOR and PST models

3.Impact of coercive and non-coercive environmental supply chain sustainability drivers on supply chain performance: mediation role of monitoring and collaboration

4.A Fuzzy Multi-Objective Mathematical Programming Model for a Closed-loop Supply Chain Network Design by Considering Environmental Factors

5.Design of Closed-Loop Supply Chain Network Considering Uncertainty in the Quality of Recycled Products and Solving It with Lp-Shape Scenario Reduction Algorithm

*Thanks for the five articles recommended by the Reviewer. They have been cited in the Introduction of this study and provide a comprehensive view for SCM study in this research. However, it seems like the article no.5. “Design of Closed-Loop Supply Chain Network Considering Uncertainty in the Quality of Recycled Products and Solving It with Lp-Shape Scenario Reduction Algorithm” has not been posted. We couldn’t find it online.

Reviewer 2 Report

Please, see attached file for comments. 

Author Response

Reviewer 2

Revision Note

Overall the article is well-written and interesting. The study seems to have been carried out in a solid and sound way, and it is presented in a clear way (although some suggestions for further clarification will follow). However, I think that the structure of the article might need some revision. My main objection is that the section called Discussion is more of a continuation of the Analysis and Results section. But in my opinion, only minor changes are needed.

Thanks for the advice from the Reviewer. The study has merged sections 4 and 5 and moved some of the internal texts.

1 Introduction

The introduction is clear and relevant.

(1) The authors could consider if, perhaps, splitting the rather long introduction into a few sections with subheadings might make it easier to navigate through the text. (This is just a thought, not a strong recommendation).

(2) Lines 40-41: I do not understand the sentence. What is meant by ”theories pedagogy”? Could it be rephrased?

(3) Line 41: ”Some restrictions” is a rather vague expression. Could some examples be given or some kind of guidance to the type of restrictions you are referring to?

(4) Line 117: 2… value of students”. Should it be value for students?

(1) Thanks for the advice from the Reviewer. This study divides Introduction into 1.1. Educational Environment and Background, 1.2. Business Management Learning, 1.3. Research Purpose.

(2) The study aimed to emphasize “theory and practice integration” and has revised the Article.

(3) Descriptions have been supplemented in the Article.

(4) Yes, the study conducted the investigation from the students’ perspective and found that they can reach the target value they want to pursue from the learning process.

2 Theoretical Background

Lines132-135: It is claimed that DGBL is more effective than traditional teaching for learning outcome – can we really know that this is true in all learning contexts and regarding all types of knowledge? Could the statement be somewhat moderated? Perhaps just change the verb “is more effective” to seems to be in this initial sentence? Later on you give good examples that show the specific contexts that have been studied so it is just the first sentence that I would suggest moderating.

* Thanks for the advice from the Reviewer. Amended the content of this study.

3 Research Methodology

(1) In 3.2 Research Procedure, lines 260-269, you mention that you performed interviews. You could add (see section 3.3) or in some other way indicate where the questions are presented so that the reader will know where to find them.

(2) Also in lines 287-289, you could refer to a section where you present the cut-off value in greater detail. In addition, I think that you need to present and describe the cut-off values before the Discussion, preferably somewhere in connection to Data analysis and Results, since you base part of your analysis on them. (I will come back to this point later on).

(1) The study added the “see the interview questions in section 3.3” in section 3.2.

(2) The study added the explanation of cut-off values in the section 3.2.

4 Data Analysis and Results

(1) Lines 325-326: The sentence starting with “Considering” is not a full sentence as I read it. Could it be rephrased? I would also rephrase “There is no question that”…, perhaps expressing instead that LOV has been shown to be a useful and reliable tool….

(2) Line 327: Since I am not at all familiar with LOV, I did not grasp if the 31 elements were identified in your study of if they come from LOV. If they come from LOV (which I think), you could add “In LOV, a total of 31…. “ . If not, you might clarify how they were identified and how they relate to LOV.

(3) Table 1 is helpful when reading the sections describing the results. Would it also be possible to include a section that summarizes the results/main findings at the end of the Analysis and Results section since there are so many results?

(4) Somewhere (perhaps in the discussion) the limited number of participants should be discussed in relation to the generalizability of the results. There are only 10 male/females for each programme and I think it is customary to mention some limitations of the study.

(1) Thanks for the advice from the Reviewer. The study rephrased a sentence in a gentle tone and adequate wording in the article.

(2) The study added the definition of values in the section 4.1. In the total of 31 elements, 9 target values come from LOV.

(3) Thanks for the advice from the Reviewer. “4.8. Summary of Findings” and a description for this section were added at article.

(4) Thanks for the advice from the Reviewer. “5.2. Limitations and Directions for Future Research” and a description for this section were added at the end of the article.

5 Discussion

(1) I do not think that the text in this section (5) is a discussion of the results. I would say that it is a second level of analysis and results based on the results presented previously. I would suggest you incorporate section 5 as a continuation of the Analysis and Results section. (You actually begin 5.1, 5.2 and 5.3 with the word “Analysis”.)

(2) Section 6 might function as a Discussion of the results – I think that is what you do there.

(3) I will now give a few comments on the analysis presented in section 5 that I hope you will find useful also if you move the text.

(4) Lines 424-430: I would recommend you to explain more clearly what the cut-off score is used for. What is cut off? (I guess it means that you took away subjects with fewer than 4 responses – but I am not sure). You could also explain why the cut off was set at 4 and what is meant by “top 1” and “top 2”. Some elaboration on Leppard et al. would be beneficial and you could also explain how to read/interpret the figures that follow. In section 3.2 you mention the Implication Matrix and the HVM but I think readers would appreciate some more guidance to understand the figures here. For example, some arrows are thick, others not, and there are different levels. Would it possible to give some short explanatory comments?

(1~3) Thanks for the advice from the Reviewer. The study has merged sections 4 and 5 and moved some of the internal texts.

(4) The study added the explanation in the section 3.2

6 Conclusion and Implications

(1) As already mentioned, I think this section is actually the Discussion (and should then become section 5 with a new heading: Discussion). Since it is a study of learning paths and tools for learning, your discussion of implications is particularly relevant.

(2) Lines 599-605: Based on your results, you recommend that digital learning software with male/female students in mind should be designed in different ways. But is there not also variation between males and between females? When do you design games just for males or females? Are not all programmes open for both male and females? Could games be designed so that they meet the needs identified for both groups? If you find these thoughts relevant, you might bring them into your discussion, if not, you could perhaps argue more strongly for the separation of male and female games.

The discussion (in section 6) is relevant and easy to follow.

(3) You could consider writing a new, short concluding section – a new section 6 – where you could try to look at your results from some distance, more from the outsider’s perspective; what can we learn from the study beyond the immediate results? Some more general comments on the contribution of the study. I think 5-10 lines would be enough.

(1) The study has merged sections 4 and 5 and moved some of the internal texts.

(2) Thanks for the advice from the Reviewer. The study provided further clarification regarding the game design. According to the economic scale and cost considerations, we believe that the game designer would not design two versions of the game due to gender. The study is more inclined to have the game designer consider all elements in a single version. Subsequently, teachers can use these designs to give different gender students a better learning experience.

(3) Thanks for the advice from the Reviewer. This research added a description for research contribution at the end of the section 5.1.

Reviewer 3 Report

The paper is very well written and enjoyable to read. Introduction, theoretical background and methodology are clearly described. Results are also clearly presented. The results show that games are an appropriate tool to allow individual learning (which I think is one of the very interesting results of this investigation. All students are playing the same simulation game, but, have different take-aways!).

However, I had difficulties to follow the aim of the study as well as the implications. In the paper, methods and results show that a) students gain different learning effects using the same simulation game depending on gender or educational background, and b) attributes of the simulation game are perceived differently by students depending on gender or educational background. These results could be more conclusive with the multiple aims of the study mentioned in the paper, a) ‘to help teachers choose the right approach and teaching method…’ (p.4), b) to ‘ discerns the differences in the path and value of students in digital game-based-learning’ (p.3), c) to ‘investigate the area-specific learning differences to develop policies for cultivating talents…’ (p. 3), d) ‘to explore system attributes of the business simulation game.. (p.3). Or is the main purpose of the study to show that students verbalize different learning paths that depend on their gender or educational background thanks to the ladder technique? What is the main objective? In line with the presented results, the overall goal of the study could be that simulation games are a beneficial tool for vocational education because they provide individual learning. Or, is the aim observing that students in vocational education learn differently because of differences in gender or educational background? I recommend prioritizing and clarifying the aim, and make it more conclusive with presented results: what is the main objective of the paper, what is the research question of the investigation, and what kind of implications can be made for which target group (vocational teachers, game designers, policy makers)?

There are also final conclusions missing on benefits for vocational education as mentioned in the introduction. Thus, I recommend bringing the aims addressed in the introduction more in line with results, discussion and implications/conclusion.

Please specify if all students of the sample were students of higher vocational education.

Furthermore, I recommend discussing limitations of the study.

Finally, I recommend a definition of learning paths as well as the understanding of values used in this paper.

Author Response

Reviewer 3

Revision Note

However, I had difficulties to follow the aim of the study as well as the implications. In the paper, methods and results show that a) students gain different learning effects using the same simulation game depending on gender or educational background, and b) attributes of the simulation game are perceived differently by students depending on gender or educational background. These results could be more conclusive with the multiple aims of the study mentioned in the paper, a) ‘to help teachers choose the right approach and teaching method…’ (p.4), b) to ‘ discerns the differences in the path and value of students in digital game-based-learning’ (p.3), c) to ‘investigate the area-specific learning differences to develop policies for cultivating talents…’ (p. 3), d) ‘to explore system attributes of the business simulation game.. (p.3). Or is the main purpose of the study to show that students verbalize different learning paths that depend on their gender or educational background thanks to the ladder technique? What is the main objective? In line with the presented results, the overall goal of the study could be that simulation games are a beneficial tool for vocational education because they provide individual learning. Or, is the aim observing that students in vocational education learn differently because of differences in gender or educational background? I recommend prioritizing and clarifying the aim, and make it more conclusive with presented results: what is the main objective of the paper, what is the research question of the investigation, and what kind of implications can be made for which target group (vocational teachers, game designers, policy makers)?

*The study simplified the multiple research aims. The research objective focuses more on Individual differences.

There are also final conclusions missing on benefits for vocational education as mentioned in the introduction. Thus, I recommend bringing the aims addressed in the introduction more in line with results, discussion and implications/conclusion.

*Thanks for the advice from the Reviewer. Discussion on the vocational course has been included at the end of the section 5.1. in this study show the contribution of this study more clearly.

Please specify if all students of the sample were students of higher vocational education.

P.7: For this reason only persons who had previous experience of playing the SCLS chosen for this study were eligible subjects.

Were the students from higher vocational education?

*Yes, all the subjects of this study are graduate school students of the university of technology, and explanations are added in the 3.3. Sampling and Data Collection section of the study.

Furthermore, I recommend discussing limitations of the study.

*Thanks for the advice from the Reviewer. “5.2. Limitations and Directions for Future Research” and a description for this section were added at the end of the article.

Finally, I recommend a definition of learning paths as well as the understanding of values used in this paper.

*The study added the definition of learning path in the last paragraph of section 2.2., and added the definition of values in the section 4.1.

Title: Individual differences in digital game-based supply chains management learning: Evidence from higher vocational education in Taiwan

l  This title sets the main aim of your study: differences in digital-game-based learning in the field of supply chain management. If this is the main aim of the study, clarify this in the paper and connect it between the introduction, results, and discussion.

Abstract: The purpose of this study was the identification and discussion of differences in learning ability by individuals studying supply chain management, in terms of the learning path and value for students of different genders and educational background.

l  This aim differs slightly from the aim mentioned above. Here, general differences in learning paths are the aim of the study, not, as mentioned above, differences in digital game-based supply chains management learning. Please clarify.

P.3: Therefore, this study first discerns the differences in the learning path and value of students of different genders and backgrounds in digital game-based learning to investigate the area-specific learning differences to develop policies for cultivating talents with inter-disciplinary integration ability in the future. For the purpose of this study, students have been separated into different groups based on two major criteria, gender and educational background, to determine the learning paths and values in both groups before a cross-analysis was performed. More specifically, the purpose of this paper can be summarized as follows: (1) To explore the system attributes of the business simulation game that students prioritize and the learning consequences and target value derived from these attributes; (2) To construct a complete picture of the value network structure that includes “system attributes—learning consequences—target value”; and (3) To compare the differences in the learning paths and values for students of different genders and educational background in a cross-analysis of the two groups.

l  In this chapter multiple aims of the study are mentioned. I recommend prioritizing aims, and clarifying goals to distinguish overall goals from research questions.

l  I also recommend writing 'To our knowledge, this study first discerns...' This is a very strong statement. Do you know the work of Hartmann and Klimmt, 2006 or Lawrence et al., 2018)?

Hartmann, T., & Klimmt, C. (2006). Gender and computer games: Exploring females’ dislikes. Journal of Computer-Mediated Communication, 11(4), 910-931. https://doi.org/10.1111/j.1083-6101.2006.00301.x

Garber Jr. L. L., Hyatt, E. M., & Boya, Ü. Ö. (2018). Constituting, testing and validating the gender learner profiles of serious game participants. The International Journal of Management Education, 16(2), 205-223. https://doi.org/10.1016/j.ijme.2018.02.005

* The study simplified the multiple research aims. The research objective focuses more on Individual differences.

* Thanks for the advice from the Reviewer. Amended the content of this study.

*Thanks for the two articles recommended by the Reviewer. They have been cited in the Introduction of this study and provide a comprehensive view for gender study in this research.

P.2: The management of a global supply chain and logistics has not only become a key aspect of the discipline, but is also one of the most important components of business management education.

in vocational education? I recommend adding some information on the question how management of global supply chain is relevant for vocational education. Is it part of curricula? Or, is this a shift to general business management education, including universities?

*The study added a case study for global supply chain management in Taiwan and other countries in the 1. Introduction and research purpose section.

P.3: On the other hand, students of different educational backgrounds also showed substantial differences in terms of their learning preferences, learning strategies and cooperative learning [29,30,31].

I recommend adding a definition of learning paths and values in the context of your investigation.

*The study added the definition of learning path in the last paragraph of 2.2. section, and added the definition of values in the 4.1. section.

P.5: This could mean that digital learning takes individual differences in students into account and can therefore ensure their learning rights and privileges [76].

I recommend referring to this statement in the discussion.

76. Sun, P. C., Cheng, H. K., & Finger, G. (2009). Critical functionalities of a successful e-learning system: An analysis from instructors’ cognitive structure toward system usage. Decision Support Systems, 48(1), 293-302.

*Thanks for the advice from the Reviewer. The study added Sun et al. (2009) in the discussion part in 4.6. HVM Analysis Result for Educational Background section.

P.7: In-depth interviews require a great deal of stamina and energy to ensure that all the valuable data could be collected while it was still fresh in the minds of the subjects.

better: 'while the data was still present to the subjects'

P.16: With regards to the differences identified in this study between the genders and by the analysis of educational background.

',' ?

* Thanks for the advice from the Reviewer. Amended the content of this study.

P.7: There is no question that LOV serves as a useful and reliable tool for the definition and conclusion of terminal values.

Such a strong statement should be based on a reference.

* Thanks for the advice from the Reviewer. The study rephrased a sentence in a gentle tone and adequate wording in the article.

P.15: Findings suggest that digital learning software designed for students of Business Administration should be developed using Scaffolding Theory as the core so that students would be able to build and establish their own game rules.

I recommend that if findings are mentioned here, cite them.

*The study cited the Scaffolding Theory proposed by Wood et al. (1976) in the article.

Round 2

Reviewer 1 Report

Accept

Author Response

Reviewer 1

Revision Note

Accept

* Thanks for accept from the Reviewer.

Reviewer 3 Report

The authors have implemented the changes very well with regard to the aim of the study. It is now more consistent. The structure has also been improved, but with a few changes it would be easier to follow the statements. Therefore, you will find my suggestions as comments in the attached PDF. I hope you can follow my suggestions.

Author Response

Reviewer 3

Revision Note

Abstract: In this study, differences in student learning paths and values were studied by examining differences in gender and educational background as a cross-analysis of the two criteria.

This sentence repeats the first sentence of the abstract. I recommend deleting it. I recommend instead: “To investigate the research question we conducted a cross-analysis of differences in gender and education background.”

* Thanks for the advice from the Reviewer. We amended all the suggestions.

Abstract: Findings from the study revealed that irrespective of gender, educational background or the cross-analysis

To me, the listing at the end of this sentence does not make entirely sence because cross-analysis is not an influential factor as gender or educational background. Cross-analysis was the method. I would not include it in this list. I recommend reducing the list to"... of gender and educational background.

1.1 Educational Environment and Background

I am not a big fan of subheadings in the introduction. However, I understand the need for it.

P.2: This helps trainees to better understand the methodologies and theories of management while allowing them to refine their practical skills through business simulation games [8]. These games create realistic simulations of real world scenarios and serve as tools that can be used to help students make more informed and better decisions [8]. Studies have shown that business simulation games offer significant benefit and help students with decision-making and improve other management-related skills and competence [13].

The transition to games seems very sudden to me. I recommend some minor changes:

'This helps trainees to better understand the methodologies and theories of management while allowing them to refine their practical skills through, for instance, business simulation games [8]. Business simulations games ...'

P.2: There are different kinds of supply chain problems in all industries. People need to face and solve them. Sustainable development of the industrial ecosystem depends on the operation of the supply chain [15]. The era of e-commerce is coming.

This information seems redundant to me. I recommend deleting it. However, in the next sentences I would include the link to sustainability:

'Global supply chain and logistic management play a crucial role for sustainable development and they are the key....'

1.3. Research Purpose

I recommend selecting a topic-related sub header such as 'Individual learning differences'.

P.4: Numerous studies have shown that when elements of quests, multimedia, interactivity and scenario simulation are incorporated, digital game-based learning (DGBL) seems to be more effective compared to a traditional teaching format in……

can

P.5: The primary aim being to identify and discern individual differences, …

of the present study being..

P.5: The primary aim being to identify and discern individual differences, as well as any learning differences, …

'...differences in playing a BSG, as well...'

3.1. Supply Chains Learning System

I my experience this could also be named: Material

3.2. Research Procedure

'Research" not necessary

4.8. Summary of Findings

5. Discussion and Implications

P.5:First, in coding results, SCLS have been identified a total of 31 elements, with 9 system attributes, 13 learning consequences and 9 target values.

Second, in content analysis results, the system attributes, learning consequences and target values emphasized in different genders, educational backgrounds and gender and educational background cross-analysis are different. The learning consequences acquisition is more concentrated for male students and more diverse for female students. The learning consequences focus by the students participating in digital learning and education is more widely. Interestingly, students from different educational backgrounds all believe that Train organizational thinking (C5) and Improve operational performance (C2) are important learning consequences.

Third, in HVM analysis results, studies found significant differences between male and female students, especially learning consequences and target values. The findings in educational background analysis, during SLCS learning, students with different educational backgrounds focus on different key points. The findings show that there is not much difference between male and female students for System attributes. There is a greater difference in Learning consequences and Target value. This demonstrates that all colleges are on a clear track to foster talents, but learning differences exist among individuals.

I recommend starting the discussion sections with this paragraph. I also don´t understand the first sentence. What do you want to say? Alternative:

'In the study, a supply chain learning system has been chosen as experimental tool. 31 elements (OF?!) have been identified containing 9 system attributes, 13 learning consequences and 9 target values. Furthermore, in content...'

Start the third paragraph with 'finally'.

5. Discussion

see above. Discussion starts now with prior paragraph

5.1. Discussion and Implications

see comments above.

P15: As established previously, this study presents a report of the results of a cross-analysis of the structure of “system attributes—learning consequences—target value” in three groups of students of both genders with different educational backgrounds who had ...

I recommend putting this paragraph to the end of the paper as a new section '7. Conclusion'.

P.15: …played SCLS. The value network and learning paths revealed in the investigation offered useful data for teachers interested in the design of digital learning courses and for the developers of relevant software and applications.

also put this to new section conclusion.

P.17: This study explores the difference in learning paths for students with different educational backgrounds and genders. It helps higher vocational schools and their teachers plan for adequate educational games and integrate them into the course to assist learners in skill development. Furthermore, past studies found out that the student's skill development and professional knowledge and the student’s confidence in himself can be used to predict his career path [116,117]. In other words, employers want to hire graduates with professional knowledge and skills. These skills must be able to be used at the workplace [118]. If the person who designed the vocational education course does not understand the learning difference among students with different educational backgrounds and genders, it is hard to design a digital learning game that meets learner’s needs and a high degree of contextualization.

This paragraph should be the second paragraph of the new section '7. Conclusion'. Please make sure to align this paragraph with the prior paragraph that I recommend for the conclusion on page 15-16.

5.2. Limitations and Directions for Future Research

6. Limitations